# InstructCoder: Empowering Language Models to Edit Code

**Qisheng Hu**[1*]    **Kaixin Li**[1*]    **Xu Zhao**[1]    **Yuxi Xie**[1]    **Tiedong Liu**[1]    **Hui Chen**[2]
**Qizhe Xie**[1†]    **Junxian He**[3†]
[1]National University of Singapore    [2]Singapore University of Technology and Design
[3]Shanghai Jiao Tong University
{qishenghu,likaixin,xu.zhao,xieyuxi,tiedong.liu}@u.nus.edu,
hui_chen@mymail.sutd.edu.sg,
junxianh@sjtu.edu.cn

## Abstract

Code editing encompasses a variety of pragmatic tasks that developers deal with daily. Despite its relevance and practical usefulness, automatic code editing remains an underexplored area in the evolution of deep learning models, partly due to data scarcity. In this work, we explore the use of large language models (LLMs) to edit code based on user instructions, covering a broad range of implicit tasks such as comment insertion, code optimization, and code refactoring. To facilitate this, we introduce InstructCoder, the first dataset designed to adapt LLMs for general-purpose code editing, containing high-diversity code-editing tasks. It consists of over 114,000 instruction-input-output triplets and covers multiple distinct code editing scenarios. The dataset is systematically expanded through an iterative process that commences with code editing data sourced from GitHub commits as seed tasks. Seed and generated tasks are used subsequently to prompt ChatGPT for more task data. Our experiments demonstrate that open-source LLMs fine-tuned on InstructCoder can edit code correctly based on users' instructions most of the time , exhibiting unprecedented code-editing performance levels on par with ChatGPT. Such results suggest that proficient instruction-finetuning can lead to significant amelioration in code-editing abilities. The dataset and the source code are available at https://github.com/qishenghu/InstructCoder.

## 1 Introduction

Developers typically engage in a cyclic routine of writing and revising code. As a crucial element, automatic code editing could potentially enhance development efficiency significantly. However, the intricacy of this task has hampered substantial progress by deep learning models. This is attributable to the fact that code editing encapsulates diverse subtasks, such as code optimization, comment insertion, and bug fixing. Each of these diverse subtasks presents distinct challenges and requires unique capabilities to solve, thereby posing considerable hurdles for modeling.

Recent development of large language models (LLMs) has made remarkable progresses in NLP, demonstrating strong few-shot and zero-shot abilities (Brown et al., 2020; Scao et al., 2022; Chowdhery et al., 2022; Ouyang et al., 2022; OpenAI, 2022; Touvron et al., 2023). Beyond text models, code LLMs have also elicited significant interest, highlighting their immense potential in code generation (Nijkamp et al., 2023a; Chen et al., 2021; Li et al., 2023). Inspired by these advancements, we explore the proficiency of LLMs in editing code based on user instructions, for instance, "add docstring to the function for clarity", "remove redundant code", or "refactor it into reusable functions".

To this end, we curate a code editing dataset, dubbed InstructCoder, for improving and evaluating code editing abilities of LLMs. InstructCoder is an instructional dataset containing diverse code-editing tasks. The dataset is primarily generated by ChatGPT (OpenAI, 2022). Specifically, we first collect and manually scrutinize git commit data from public repositories on GitHub as the seed code editing tasks, then we utilize the seed data to prompt ChatGPT to generate new instructions and input-output pairs respectively, where a scenario (e.g. web development) is randomly sampled from a list of scenarios and specified to ensure diversity of the data. This process resembles the Self-Instruct (Wang et al., 2022a) and Alpaca (Taori et al., 2023) frameworks.

By innovatively incorporating scenarios during

---

*Equal contribution. Ordering is determined by dice rolling.

†Equal advising. Ordering is determined by dice rolling.

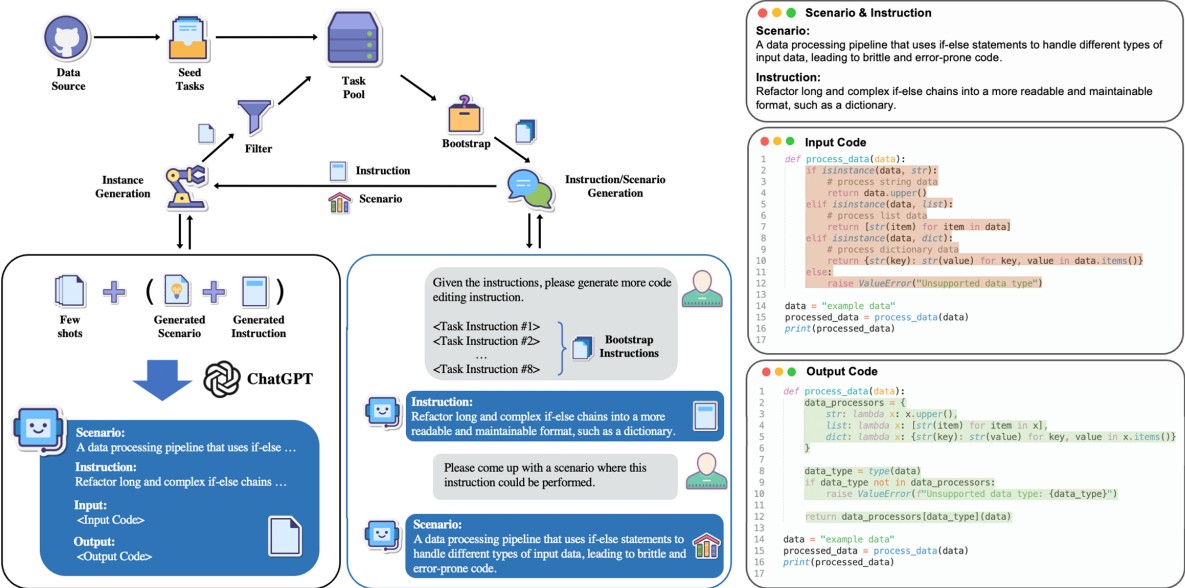

Figure 1: Data collection pipeline of InstructCoder (left) and an qualitaive example from the dataset (right, best viewed with zoom). Initial seed tasks are selected from GitHub commits, and inspire ChatGPT to generate new instructions. Plausible scenarios where the filtered instructions may be used are then generated. Finally, corresponding code input and output are obtained conditioned on both the instruction and scenario. High-quality samples are manually selected and recurrently added to the task pool for further generation.

the generation process, our approach ensures that the code-editing instances in the InstructCoder dataset are diverse and relevant to real-world programming situations. This approach enables ChatGPT to synthesize more diverse input-output code snippets in terms of variable naming and functionality given the code-editing instructions and scenarios, resulting in a robust dataset for instruction finetuning in the code editing domain. After proper deduplication and postprocessing, we retain over 114,000 samples in the dataset.

Our empirical studies reveal that LLMs display notable gains in code editing abilities post finetuning on InstructCoder. The largest model used in the experiment, LLaMA-33B (Touvron et al., 2023), performs on-par with ChatGPT, achieving an edit accuracy of 89.3% and 76.3% as evaluated by GPT-4 (OpenAI, 2023) and humans respectively. Further findings signify that edit accuracy improves log-linearly with data scale.

## 2 Related Work

### 2.1 Instruction Finetuning Datasets

Previous studies have concluded that instruction finetuning LLMs on a diverse collection of in-

structional tasks can further improve the ability of LLMs to generalize well on unseen tasks (Ouyang et al., 2022; Mishra et al., 2022; Wei et al., 2022; Chung et al., 2022; Wang et al., 2023c). To support these tasks, datasets consisting of a large number of code snippets with corresponding annotations are necessary. These instruction can be reformulated from existing datasets (Aribandi et al., 2022; Wei et al., 2022; Mishra et al., 2022; Longpre et al., 2023), or human-written with crowd-sourcing efforts (Ouyang et al., 2022; Wang et al., 2022b). Machine generation of instruction data has also been explored to reduce human labour (Wang et al., 2022a; Honovich et al., 2022; Taori et al., 2023; Xue et al., 2023). Despite the presence of elevated noise levels within the data, its effectiveness has been identified.

### 2.2 Code Synthesis

Code generation is an extensively studied area. Language models pretrained on large collections of code have demonstrated strong abilities in a variety of programming tasks. A number of general LLMs gain code generation abilities due to the mixture of code in the pretraining corpus (e.g. The Pile (Gao

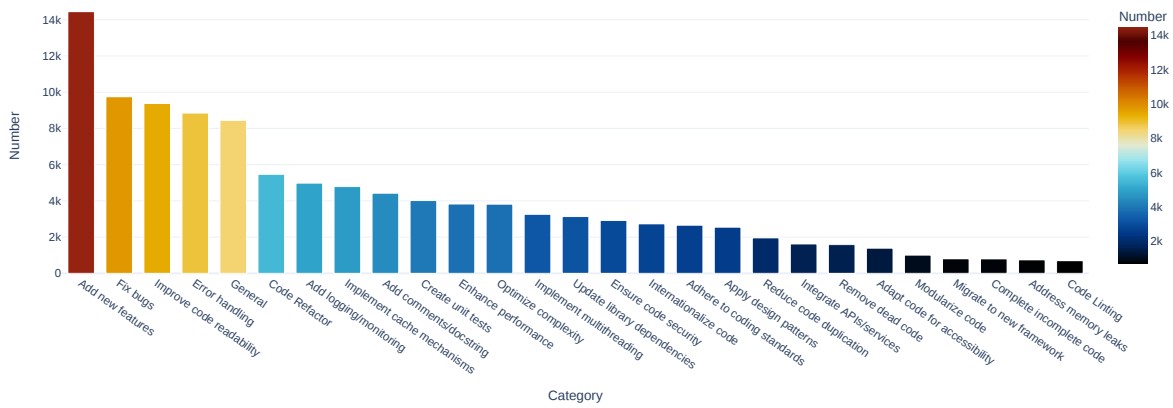

Figure 2: Distribution of code edit intent categories.

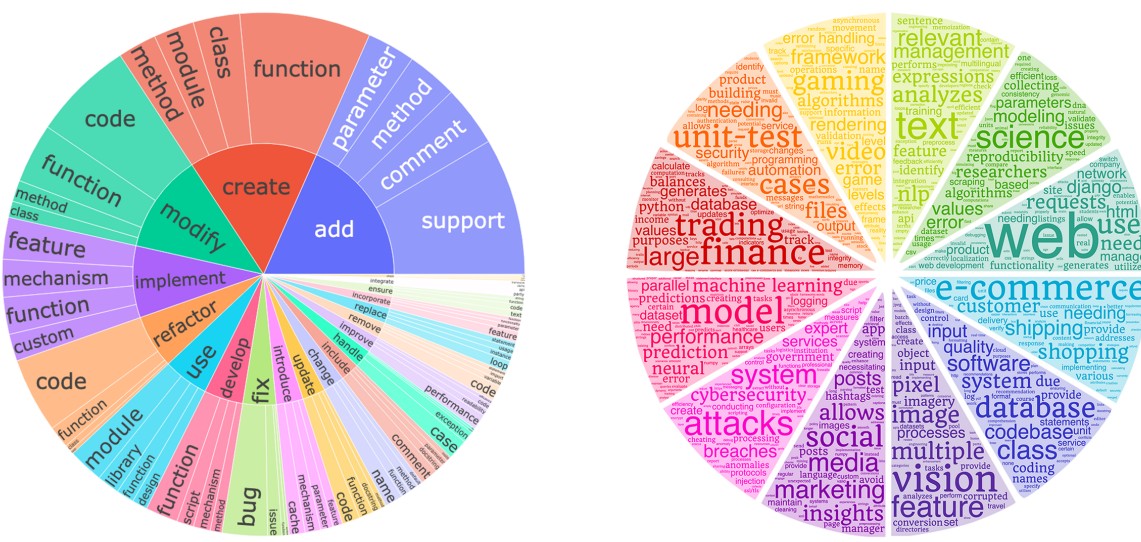

(a) The top 20 most common root verbs with each top 4 noun objects in the instructions. Instructions with other infrequent root verbs takes up 25%.

(b) Wordcloud of scenario domains. Each sector with different color corresponds to a different scenario domain. Each domain is a cluster of similar scenarios.

Figure 3: Visualizations of InstructCoder data. Best viewed in zoom.

et al., 2020)), such as GPT-3 (Brown et al., 2020), ChatGPT, GPT-4 (OpenAI, 2023), LLaMA (Touvron et al., 2023), BLOOM (Scao et al., 2022), GPT-NeoX (Black et al., 2022), and Pythia (Biderman et al., 2023). LLMs specifically trained on code and optimized for code generation are also studied, e.g. CodeX (Chen et al., 2021), CodeGen (Nijkamp et al., 2023b), CodeGeeX (Zheng et al., 2023) and StarCoder (Li et al., 2023). These models all adopt the decoder-only transformer architecture, but differ in size and specific model design (e.g. positional embedding, norm layer place-

ment) as well as the selection and preprocessing of pretraining corpus.

On the other hand, relatively fewer literature addresses the objective of code editing. Previous works focus on a subset of code editing tasks, such as code infilling (Fried et al., 2023) and debugging (Just et al., 2014; Tarlow et al., 2020; Ding et al., 2020). The PIE (Madaan et al., 2023) dataset is a concurrent work most relevant to ours, which focuses on speeding up programs. Other works (Yin et al., 2018; Wei et al., 2023; Chakraborty et al., 2020) can not accept natural

language as edit intentions, rendering them less user-friendly.

Nevertheless, datasets particularly tailored for general-purpose code editing are absent. To fill this gap, we introduce InstructCoder, a novel dataset aimed at further advancing the capabilities of code editing with LLMs.

## 3 InstructCoder Dataset Collection

To generate instructional data for code editing, we employed a method based on Self-Instruct (Wang et al., 2022a), which expands instruction finetuning data by bootstrapping off language model generation. The methodology of generating data with LLMs requires minimal human-labeled data as seed tasks while maintaining the quality and relevance of the tasks in dataset. Through an iterative process of generating instructions and refining them with deduplication, we create a dataset of a wide range of code-editing tasks. Figure 1 illustrates the data collection pipeline of InstructCoder.

### 3.1 Seed Data Collection

GitHub is a code hosting platform whose version control service naturally records code edits with commits, which can be converted to instructions. The repositories on GitHub provide diverse data with human-generated quality. However, the data cannot be directly utilized. First, commit messages are mostly brief and resultant, missing detailed descriptions. Furthermore, they can be imprecise or even absent. Second, commits can be huge involving multiple files, which is beyond the scope of this work. In light of this, we direct our attention towards LLMs as a means to generate data, instead of the direct utilization of collected data.

Initially, raw github commit data were collated through BigQuery.[1] The task instructions were derived from the commit message, while the input and output corresponded to the code version before/after the commits. We came across many imprecise or emotionally charged commit messages. To convert commit messages to proper instructions, we employed Codex (Chen et al., 2021) to clarify the changes made between versions and improve the commit messages, resulting in more precise and informative instructions. A total of 768 seed tasks

were processed from the commit data through manual efforts. 634 tasks were used for self-instruct purposes while 134 tasks reserved for evaluation.

In addition to GitHub commit data, we leverage high-quality generated samples for seed tasks. With manual inspection, we compiled a batch of 592 high-quality samples as additional seed tasks. This set of seed data cover a wide range of code-editing scenarios and forms the very basis on which InstructCoder is created, ensuring that the tasks are rooted in plausible real-world code-editing cases.

### 3.2 Instruction Bootstrapping

Self-Instruct (Wang et al., 2022a) serves as an effective automated framework for instruction data generation. It works by iterative bootstrapping off LLM's generation, presenting a way to enrich the instructional dataset while maintaining task quality and relevance from a small set of human-evaluated seed tasks. We leveraged a similar approach to generate diverse code editing instructional data. In each iteration, seven seed task instructions and one ChatGPT-generated task instruction are sampled and combined in a few-shot manner to prompt ChatGPT for more instructions. To generate more diverse and practically applicable instructions, we also generate tasks across multiple sub-domains by specifying the editing intent in the prompt provided. Relevant prompt used can be found in Table 3 in Appendix A.

### 3.3 Scenario-conditional Generation

We originally found many generated samples share similar codebases and variable names despite different instructions and few-shot examples provided. Such similarity could largely diminish the dataset's research value. Empirical analysis suggests the issue could be attributed to LLM generating general codebases for input/output snippets when insufficient context provided. To mitigate this, we introduce scenarios to input/output generation. As an illustration of the effects of scenario generation, we present some examples in Figure 7,8,9 in Appendix B, where we observe that instances generated with scenario demonstrate higher quality in terms of richer context and code structure compared to those without.

For each generated instruction, we first prompt ChatGPT to generate practical events as "real-

---

[1] https://cloud.google.com/bigquery

world" scenarios where the editing instruction could be performed, and randomly selected one for instance generation in the next step. Subsequently, the LLM is instructed to generate samples that correspond with the instruction and scenario, ensuring the codebases and variable names are appropriate. The prompt used can be found in Table 3 in Appendix A.

By incorporating scenario-conditional generation, the resulting samples exhibit increased variability in regards to codebases and variable naming, thus augmenting the diversity of InstructCoder.

## 3.4 Postprocessing

Following Self-Instruct (Wang et al., 2022a), deduplication is applied on the generated instructions to remove instructions that have a ROUGE-L (Lin, 2004) overlap score larger than 0.7 with the existing instructions. We also employ MinHash with Locality Sensitive Hashing (LSH) indexing using datasketch[2] to remove instances with an input code Jaccard similarity greater than 0.75, in order to deduplicate at the code level. More heuristic rules were used to clean the generated data. With postprocessing, we achieved a high level of effectiveness in eliminating erroneous and redundant data.

We kept 95% of the dataset as the train set and assigned 5% of the dataset as the validation set. The test set is built with held-out seed samples from real GitHub data to better reflect the real-world edit cases. Since commit messages from GitHub code edits are noisy, we conducted manual quality filtering. As a result, InstructCoder consists of 108391 training samples, 5708 validation samples and 134 test samples.

## 4 Data Analysis

We analyze InstructCoder in terms of 1) diversity, 2) complexity, and 3) correctness. We provide distribution and complexity analyses of the task instances. Finally, we demonstrate through human investigation that our data is highly reliable.

## 4.1 Statistic Overview

InstructCoder comprises over 114 thousand code editing instructions, each paired with an input/output instance. The token length distribution of input/output can be viewed in Figure 4 and Table

[2]http://ekzhu.com/datasketch/

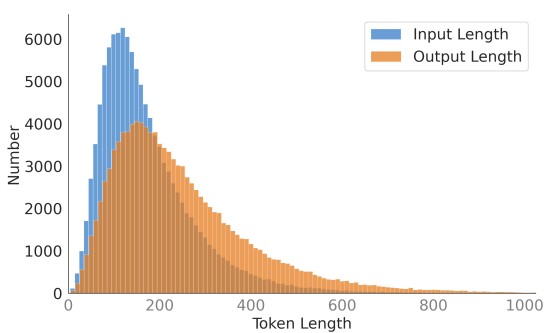

Figure 4: Token length distribution of InstructCoder

4 in Appendix C. Most of the data falls within a reasonable range in terms of length, while there are also some extreme values that reflect the breadth of our dataset.

## 4.2 Instruction Diversity

To explore the diversity of tasks in InstructCoder and their practical applicability, we present various instruction intents i.e. *what* the code edits intend to accomplish, and instruction verbs, i.e. *how* the code edit is accomplished.

**Instruction Intents.** We asked ChatGPT to classify the types of code edits in our dataset, and manually identified 27 empirical genres. Figure 2 shows the distribution of the code edit intent categories in InstructCoder, which include adding functionalities, optimizing code, improving readability, etc. These objectives underscore the extensive range of InstructCoder.

**Instruction Verbs.** The diversity of instruction verbs is also portrayed in Figure 3a. We demonstrate the top-20 root verbs and their top-4 direct nouns both ranked by frequency. While a great portion of the instructions can be roughly clustered as *creation* (e.g. "add", "implement", "creat") and *modification* (e.g. "modify", "replace", "change"), InstructCoder presents a long-tail distribution with less common verbs other than the top-20 taking up 25.0% percentage. This demonstrates that the dataset contains a wide spectrum of instructions.

## 4.3 Scenario Diversity

InstructCoder is designed to cover a wide range of scenarios. Each instruction is prompted to generate different scenarios where the editing instruction

could be performed. This approach ensures that the generated samples exhibit greater diversity in terms of codebases and contexts. A wordcloud is provided to show some of the scenario domains in our dataset, as illustrated in Figure 3b, with each sector referring to a different domain. The diversity of the dataset is emphasized by the presence of a wide range of domains such as image processing, web development, and cybersecurity.

## 4.4 Complexity

We reflect the complexity of a code edit task by the number of differing lines and its ratio in the input/output pair, which are defined as:

$$n_{diff} = |I \cup O \setminus I \cap O|, \tag{1}$$

$$r_{diff} = \frac{n_{diff}}{|I \cup O|}, \tag{2}$$

where $I$ and $O$ are sets of input/output code with single lines as elements. We measure the differing lines of a code-editing task instance using the Python library *difflib*.[3] We found that the average number of differing lines in InstructCoder is 11.9 and the average ratio is 0.52. These values suggest a fairly acceptable level of complexity, indicating that the dataset is neither too easy nor too hard. *InstructCoder* strikes a balance in terms of complexity, making it well-suited for finetuning and evaluating LLMs in a wide range of code editing tasks. Figure 10 in Appendix C illustrates the distribution of the number of differing lines.

## 4.5 Correctness

We further randomly sample 200 instances and invite three co-authors to evaluate the instances based on two criteria: the validity of the instruction and the correctness of the instances. The validity assessment focuses on deciding if the instructions clearly exhibit editing intent and are appropriate for code editing. The correctness evaluation examines if the input-output pairs reflect the changes specified by the instructions.

The results in Table 1 indicate that most instructions in the InstructCoder dataset are valid. A few instances exhibit noise and occasional failure to follow the instruction, but overall high correctness is achieved. Out of the 200 evaluated instances, 180

---

[3] https://docs.python.org/3/library/difflib.html

| Question | Pass |
|---|---|
| `Determine if the instruction is valid.` | 97% |
| `Is the output an acceptable edited code response to the instruction and input?` | 90% |

Table 1: Quality check questions and results on a randomly sampled subset with 200 data points.

were successfully solved, showcasing the overall quality and reliability of InstructCoder.

# 5 Experiments

## 5.1 Setup

**Training.** We experiment with two families of open-source language models: LLaMA (7B, 13B, 33B) (Touvron et al., 2023) and BLOOM (560M, 3B, 7B) (Scao et al., 2022). LLaMA is a series of large language models with parameter counts ranging from 7B to 65B, and pretrained with an excessive amount of tokens, wherein code takes up approximately 4.5%. BLOOM is a multilingual LLM capable of generating human-like outputs in 46 languages and 13 programming languages. A full finetuning which updates all the parameters in an LLM can be computationally expensive. Instead, we adopt LoRA (Hu et al., 2022), a parameter-efficient finetuning method which optimizes an approximated low-rank delta matrix of the fully-connected layers. Though the number of parameters updated in LoRA is typically several magnitudes lower than that of the full model, many works have demonstrated its effectiveness comparable to full finetuning (Hu et al., 2022; Wang et al., 2023a). In this way we could finetune a 33B model in a single A100-80GB GPU card. Across all our experiments, LoRA is applied on the query, key, value and output transform weights of the Transformer architecture (Vaswani et al., 2017). All hyperparameters can be found in Table 5 in Appendix D.

**Baselines.** We select zero-shot ChatGPT (OpenAI, 2022) as a strong baseline. We also include open-source models, LLaMA (Touvron et al., 2023) and Alpaca (Taori et al., 2023), and report their zero-shot and one-shot performance. We do not experiment on k-shot with larger k, because these prompts take up too many tokens.

Concurrent to our work, CodeAlpaca[4] is a popular dataset generated with the pipeline of Alpaca (Taori et al., 2023). Its seed data is replaced by hand-written easy instructions with short programs. We finetune LLaMA models with CodeAlpaca and compare the results.

## 5.2 Metrics

Evaluating the accuracy of code edits presents a complex challenge due to the potential for incomplete code snippets and the existence of multiple valid modifications. Evaluating correctness using conventional metrics proves arduous, hence our reliance on human evaluation. Each sample is annotated by three examiners, and the average accuracy is reported. We also endeavored to prompt GPT-4 (OpenAI, 2023) in inspecting the modifications.

**Human Scoring.** We establish a rule indicating three scoring levels: *correct*, *partial* and *wrong*. To ensure impartiality, output samples from different models are shuffled and each is evaluated by three co-authors using a tool that guarantees the anonymity of the models was used. The edit is assigned *correct* if it correctly reflects the instruction demands and *wrong* if it fails to follow the instruction. We introduce a *partial* class to contain subtle situations where the edit is correct but unexpected modifications are made, such as removal of comments or redundant continuous generation.

**GPT-4 (OpenAI, 2023) Evaluation.** We leverage GPT-4 as an automatic evaluator to alleviate the need of human effort and ensure fair evaluation. Using LLMs as generation evaluators has been demonstrated effective in NLG tasks (Liu et al., 2023; Wang et al., 2023b; Fu et al., 2023), and especially in code generation (Zhuo, 2023). We prompt GPT-4 to evaluate if the code edit is an acceptable response to the input and instruction. The prompts can be found in Table 3 in Appendix A.

## 6 Results

### 6.1 Finetuning Efficacy with InstructCoder

Table 2 provides a comprehensive comparison across models finetuned with InstructCoder and the baselines. We leave the discussion of the validity of

| Model | Size | Accuracy (%) |
|---|---|---|
| *Baselines* | | |
| ChatGPT (0-shot) | - | 90.5 |
| BLOOM (0-shot) | 3B | 3.0 |
| BLOOM (1-shot) | 3B | 3.0 |
| BLOOM (0-shot) | 7B | 5.2 |
| BLOOM (1-shot) | 7B | 11.7 |
| LLaMA (0-shot) | 7B | 12.4 |
| LLaMA (1-shot) | 7B | 14.2 |
| LLaMA (0-shot) | 13B | 18.7 |
| LLaMA (1-shot) | 13B | 25.6 |
| LLaMA (0-shot) | 33B | 24.1 |
| LLaMA (1-shot) | 33B | 54.5 |
| *Finetuned with Alpaca Dataset* | | |
| | 7B | 39.3 |
| LLaMA | 13B | 55.2 |
| | 33B | 70.6 |
| *Finetuned with CodeAlpaca Dataset* | | |
| LLaMA | 13B | 48.5 |
| | 33B | 74.6 |
| *Finetuned with CodeInstruct Dataset* | | |
| | 560M | 20.9 |
| BLOOM | 3B | 51.2 |
| | 7B | 56.2 |
| | 7B | 69.2 |
| LLaMA | 13B | 75.9 |
| | 33B | **89.3** |

Table 2: Accuracy evaluated by GPT-4.

using GPT-4 as an evaluator and human scoring results in Appendix E. The average of three runs was taken for each score. We also showcase human-evaluated model performance finetuned with InstructCoder in Table 6. While low accuracy are observed in plain open-source models and only marginal improvement is achieved through few-shot prompting, finetuning with InstructCoder significantly boost the accuracy, suggesting the effectiveness of efficient instruction finetuning with machine-generatied code edit pairs.

It is noteworthy that our largest finetuned LLaMA-33B exhibits a performance comparable with the strong baseline ChatGPT on the test set. Some qualitative results are shown in Appendix F. Despite the noise present in the data points collected through git-diff,[5] which might entail incomplete contextual information and some disparity in code structure, the finetuned LLaMA-33B achieves an accuracy of 89.3% under GPT-4 evaluation, with a 65% increase over its plain counterpart.

The ability of the underlying LLM also serves as a significant determinant in the code-editing capacity. While enhancements are evident across

---

[4]https://github.com/sahil280114/codealpaca

[5]https://git-scm.com/docs/git-diff

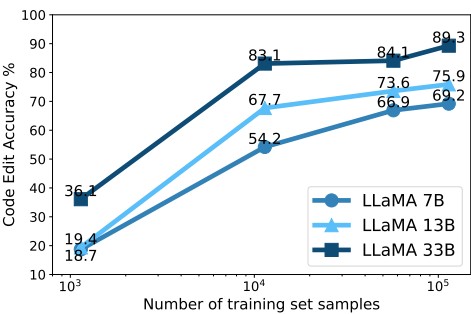

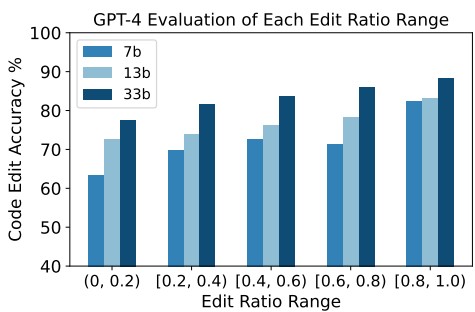

Figure 5: Data scaling performance of InstructCoder on LLaMA evaluated by GPT-4, using 1%, 10%, 50% and 100% training data.

Figure 6: GPT-4 evaluation results at different edit ratios on 2000 validation samples.

all finetuned models, the LLaMA models exhibit superior accuracies when compared to BLOOM models of comparable sizes.

The models finetuned with CodeAlpaca exhibit unsatisfactory results. The 13B model exhibits a mere 48.5% accuracy, a markedly inferior performance compared to the finetuning with Instruct-Coder, and even lower than that of Alpaca. In the case of the 33B model, CodeAlpaca surpasses Alpaca in performance; however, it remains substantially worse than InstructCoder. This finding validates our methodology of employing GitHub seed data to produce a more challenging and diverse dataset. The observation suggests the presence of a considerable domain gap between CodeAlpaca and authentic real-world test data, rendering CodeAlpaca suboptimal, though the phenomenon can be partially alleviated by scaling up model size.

## 6.2 Dataset Scaling

InstructCoder has a scale considerably smaller than what LLMs are typically pretrained on. In order to ascertain the sufficiency of this scale, we conducted an experiment wherein we fine-tuned the LLaMA family of models using varying proportions (1%, 10%, 50%, and 100%) of the dataset. The data subsets corresponding to smaller proportions are guaranteed to be encompassed within the larger data subsets. The results are shown in Figure 5.

The identified trend demonstrates a positive correlation between the model's accuracy and the scale of the training set. However, this relationship exhibits diminishing returns as the dataset size continues to expand. Utilizing just 10% of the data brings significant increase and surpass the corresponding zero-shot and one-shot accuracies with-

out finetuning (see Table 2) by considerable margins. With over 10% training data, larger models demonstrates superior performance than smaller models trained with full data. except for LLaMA-13B@10% and LLaMA-7B@100%. While we empirically observed that the training time grows approximate linearly with parameter count in our experiments, the results reveals that larger models should be preferred with limited training compute budget.

## 6.3 Edit Ratios

Figure 6 shows the accuracy of finetuned LLaMA models across five levels of edit ratio. Larger models consistently outperforms smaller ones within each bin. Interestingly, the accuracy of models' edit is generally lower as the edit ratio decreases. One plausible reason is that, as the fine-tuning loss is the average of cross-entropy on the label tokens, a shortcut of copying the inputs is easily learnt by the model to achieve a fairly low loss value, especially when the absolute number of modifications is small. Our observations indicate that this issue can be alleviated by scaling up the models. Larger models perform better in capturing subtle differences in low edit ratio cases.

## 7 Conclusion

We introduce InstructCoder, the first instruction tuning dataset for general-purpose code-editing tasks. The dataset comprises generations of Large Language Models, where real GitHub commits serve as seed tasks to guide the generation process. A scenario-conditional approach is introduced to ensure both diversity and high quality of the data. Our experiments show that with computationally

lightweight parameter-efficient finetuning, open-source models can gain huge improvements and even yield ChatGPT-like performance. We also reveal that the LLM base model and the scale of finetuning data are both profound factors of code-editing ability. We hope the dataset can benefit and inspire more research in this area towards building more powerful coding models.

## Limitations

While we chose genuine github commits as the source of our seed tasks, the data produced may still exhibit biases that deviate from real-world application. Moreover, our approach did not encompass code changes involving cross-files contexts, which might be the common case in development. We hope to explore these aspects further and incorporate additional programming languages in our future research.

## Ethics Statement

This research paper adheres to the ethical guidelines and principles set forth by the Conference on Empirical Methods in Natural Language Processing (EMNLP) and the wider scientific community. All real-world data were collected only from public repositories from GitHub.

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

## A Prompts

The prompts we used in our data collection and experiments are listed in Table 3.

| Stage | Prompt |
|---|---|
| Instruction Generation | Given the existing instructions, please generate a list of diverse python code editing instructions. The new instructions should address diverse editing tasks. Please ensure that the instructions are clear and diverse. Include any relevant variable names in the instruction. |
| Scenario Generation | Given a python code editing task, please come up with 10 diverse scenarios concise description where this python code editing task could be performed or come from. |
| Instance Generation | Given python code editing task instructions and their scenarios where the task instruction could be used, you need to come up with examples for the following code editing tasks. You need to generate input and output code pair and make sure your variable names are suitable for the scenario. The input code is related to the task instruction, but must NOT meet the task requirements. The output code fulfills the task requirements based on input code. |
| GPT4 Evaluation | Given a code editing instruction, please determine if the output is an acceptable edited code response to the instruction and input? Give "Yes" or "No". |

Table 3: Prompts used in this work.

# B Qualitative Examples of Scenario-Conditional Generation

Three comparisons are presented, each showing instances that were generated with or without the inclusion of a scenario.

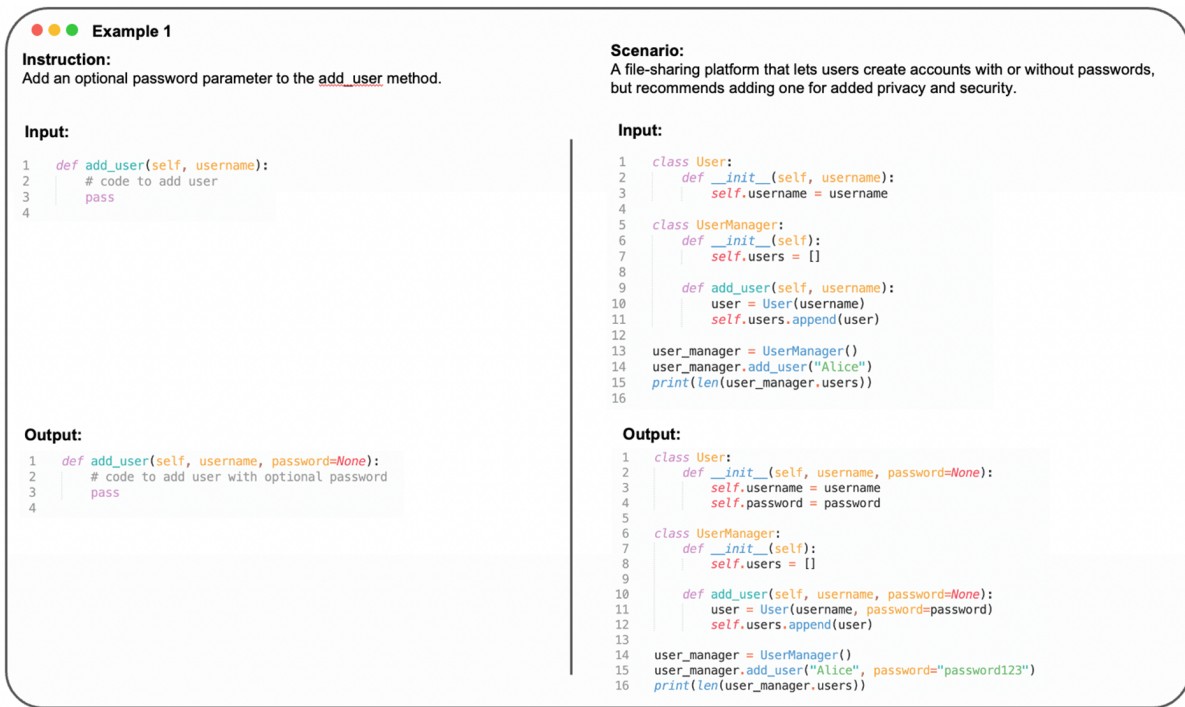

Figure 7: Example instance #1 generated without scenario (Left) and with scenario (Right)

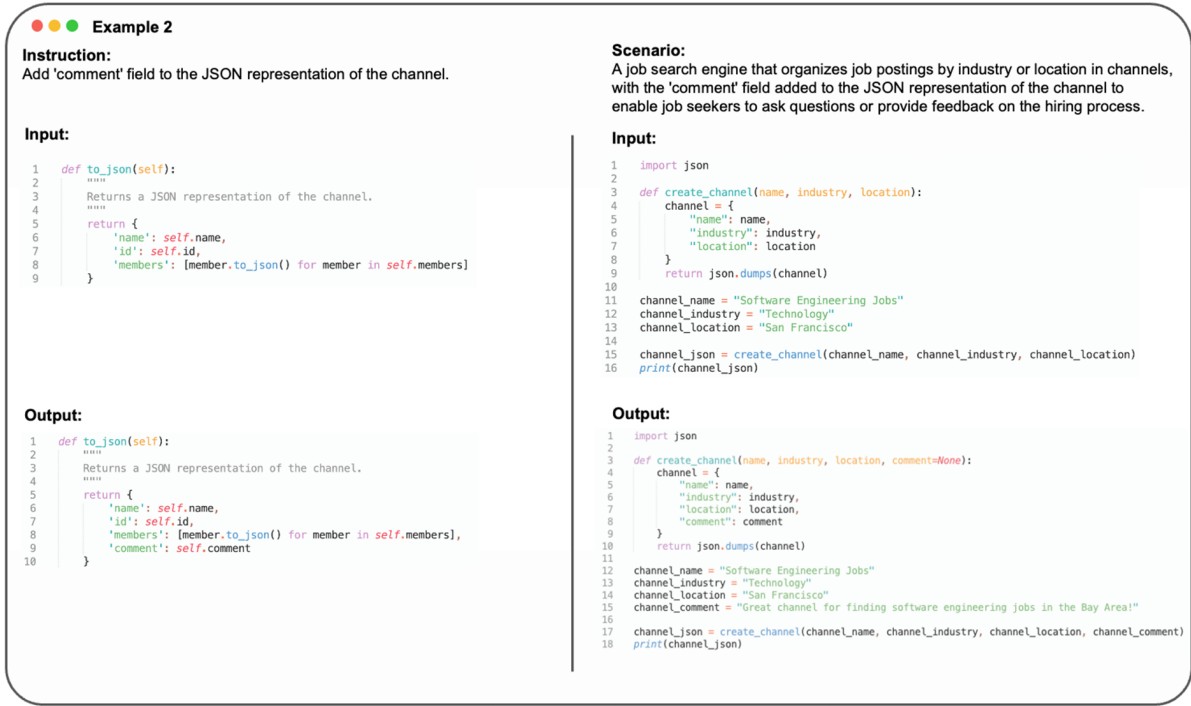

Figure 8: Example instance #2 generated without scenario (Left) and with scenario (Right)

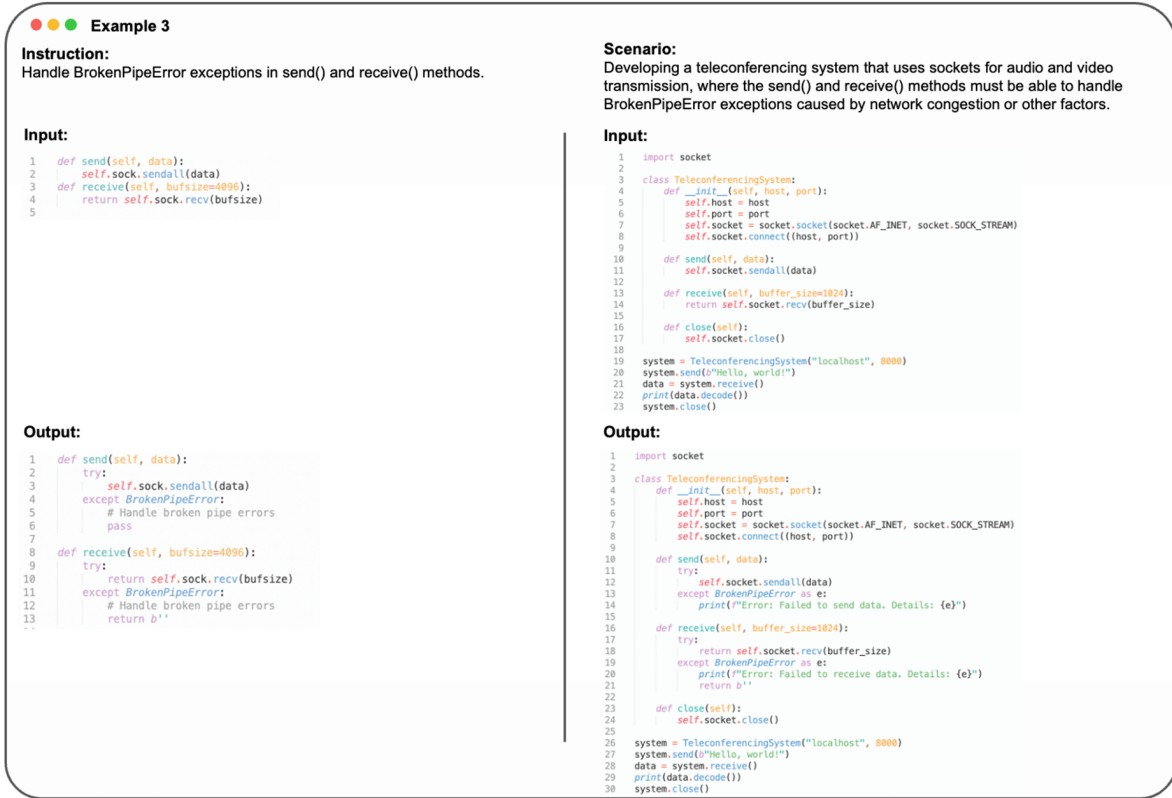

Figure 9: Example instance #3 generated without scenario (Left) and with scenario (Right)

## C Additional statistics of InstructCoder

| Token Length | Instruction | Input | Output |
|---|---|---|---|
| mean | 21.85 | 172.03 | 248.43 |
| 25% | 17 | 99 | 138 |
| 50% | 21 | 147 | 213 |
| 75% | 26 | 218 | 321 |
| min | 3 | 10 | 10 |
| max | 116 | 1019 | 1024 |

Table 4: Token length statistics using the LLaMA (Touvron et al., 2023) tokenizer.

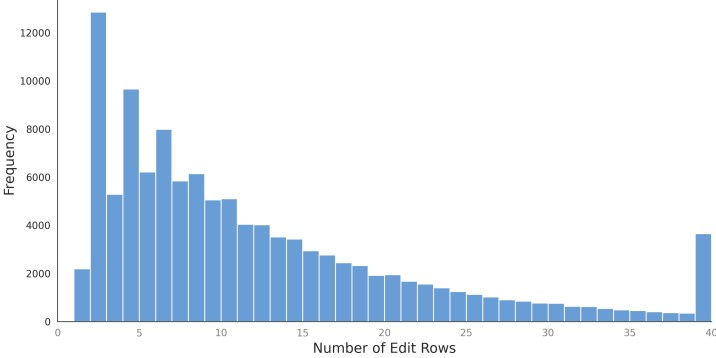

Figure 10: Edit rows distribution of InstructCoder (Number greater than 40 are aggregated as the last bin.)

## D   Hyperparameters

The hyperparameters used in all finetuning experiments is listed in Table 5.

| Hyperparameter | Value |
| --- | --- |
| learning rate | 0.0003 |
| batch size | 128 |
| epochs | 3 |
| max sentence length | 1024 |
| lora rank | 16 |
| lora dropout | 0.05 |
| lora modules | key, query, value, output |

Table 5: Hyperparameters used for finetuning language models.

# E   Human Evaluation Results and Analysis on Human Alignment of GPT-4

Using the scoring standard described in 5.2, we conducted human evaluation on the test set code edits sampled from ChatGPT and LLaMA fine-tuned on InstructCoder. The results are provided in Table 6. The results is compared to the evaluation results of GPT-4. When considering the partial type of human scoring as correct, our observations reveal an average consistency ratio of 68.2%, and on the largest evaluated model, LLaMA-33B, the value rises to 78.4%. This renders GPT-4 evaluation as an acceptable method for evaluating the correctness of code edit tasks.

| Model | Correct | Partial | Wrong |
|---|---|---|---|
| ChatGPT | 79.3 | 10.4 | 10.3 |
| LLaMA-7B | 54.1 | 8.1 | 37.8 |
| LLaMA-13B | 69.6 | 5.2 | 25.2 |
| LLaMA-33B | 76.3 | 8.1 | 15.6 |

Table 6: Human evaluation results of LLaMA models finetuned with CodeInstruct on the test set collected from GitHub. *Correct* means the edit correctly reflect the task demand. *Partial* means the edit is mostly correct, but with minor unexpected modifications. *Wrong* indicates non-acceptable edit.

# F   Qualitative Examples Generated by Finetuned LLaMA-33B

We demonstrate some qualitative example response generated by finetuned LLaMA-33B.

Figure 11: Qualitative examples generated by finetuned LLaMA-33B

## G Data Filtering Process

The detailed process of filtering the dataset is listed below:

- We selected github repos with over 100 stars to ensure the overall quality. We only utilized repos with permissive licenses (MIT, Apache-2.0, GPL-3.0, GPL-2.0, BSD-2.0, BSD-3.0, LGPL-2.1, LGPL-3.0, AGPL-3.0).

- We kept commits in which only one single .py file was changed. Using git-diff, we identified and preserved commits where only one code block was changed.

- We discarded commits with single-word or empty commit messages.

- We removed commits with over 100 edited rows.

Manual:

- We discarded rare commits containing inappropriate language.

- We discarded commits where the change in the source code does not match the commit message.

- We filtered out project-specific adjustments that lack sufficient context.

- We utilized Codex (Chen et al., 2021) to rewrite ambiguous commit messages, enhancing the clarity of the intended code edits.

