# OpenReview forum: "InstructCoder: Empowering Language Models to Edit Code"
_EMNLP/2023/Conference — Submitted to EMNLP 2023_

### Official Review · Reviewer_YC8o · 2023-07-30

**Soundness:** 3

**Excitement:**

4: Strong: This paper deepens the understanding of some phenomenon or lowers the barriers to an existing research direction.

**Missing References:**

- CodeReviewer ("Automating Code Review Activities by Large-Scale Pre-training") [https://arxiv.org/abs/2203.09095]
- CoditT5: Pretraining for Source Code and Natural Language Editing [https://arxiv.org/abs/2208.05446]
- On Multi-Modal Learning of Editing Source Code [https://arxiv.org/pdf/2108.06645.pdf]
- Using Developer Discussions to Guide Fixing Bugs in Software [https://arxiv.org/abs/2211.06335]
- CODIT: Code Editing with Tree-Based Neural Models [https://arxiv.org/pdf/1810.00314.pdf]
- Learning to Represent Edits [https://arxiv.org/abs/1810.13337]

**Paper Topic And Main Contributions:**

This paper introduces a new dataset, CodeInstruct, consisting of 114K (NL instruction, Python code input, Python code output) triplets encompassing diverse code-editing tasks. To construct this dataset, they first gather 768 seed tasks from GitHub commits data, for which they form the NL instructions by rewriting commit messages using Codex. Then, following the Self-Instruct framework, they iteratively generate new examples by prompting ChatGPT in a few-shot manner, with seven seed task instructions and one high-quality ChatGPT-generated instruction. Furthermore, for more diversity, they also include a scenario in the prompt, where scenarios are also generated by ChatGPT. In experiments, they fine-tune (using LoRA) open-source LLMs (LLaMa and BLOOM) on CodeInstruct. They perform evaluation on 134 seed examples through manual inspection and GPT-4, through which they show that their dataset can be used to fine-tune open-source LLMs to achieve comparable performance to ChatGPT.

**Questions For The Authors:**

A) Please address my first point above about using GitHub data directly or the CodeReviewer dataset.

B) Please provide further clarity on the evaluation method, based on the concerns I raised above.

C) One difference between code generation and code editing is that in editing, large portions of the input often get preserved. Have you explored output examples in some "diff" format which only requires generating the lines that actually changed?

D) For the seed data that was collected from GitHub, did you ensure that you used only license-permissive repositories? If not, using this dataset for anything beyond research purposes is likely not allowed, and this must be made clear. Please clarify this point.

**Reasons To Accept:**

- Code editing is a very important capability that has been less explored in the GPT-era (though there is extensive work in code editing in the past, see "Missing References"), so it is nice to see work that is focused on this.
- The CodeInstruct dataset offers a nice training set that can be used for further research.
- Incorporating the notion of "scenarios" when building the dataset is quite clever and novel.
- Establishing that the gap between ChatGPT and open-source LLMs with respect to code editing can almost be closed with fine-tuning on dataset generated using prompt-based data generation techniques is interesting and may have implications for further research.

**Reasons To Reject:**

- By comparing against Alpaca and CodeAlpaca, the authors demonstrate that their data generation technique is superior to other prompt-based automatic data generation techniques for this task. However, it is not clear whether such techniques are better than just directly using human-written examples from GitHub. In Lines 163-175, the authors claim that GitHub commits are too noisy to use directly; however, this is not empirically validated. GitHub serves as an extremely large data source, and given that one of the findings of this paper is that the scale of the data is a profound factor of code-editing ability, it is important to understand whether the scale of the data reduces the impact of noise. Moreover, commit messages are not the only source of NL instructions from GitHub. Another source is pull request comments (and the corresponding code edits). In fact, there is already a large-scale benchmark for this: CodeReviewer (see missing references). As a point of reference, it would be important to understand what the effect is of fine-tuning open-source LLMs on CodeReviewer, and how this compares to fine-tuning on CodeInstruct.
- The evaluation is limited and weak. First, the test set entails only 134 examples which are manually curated by the authors of the paper, and it appears that they are all in Python. Next, the main form of evaluation is prompting GPT-4 to judge the correctness of model predictions. While this has been explored for other tasks, this has not been established as a valid evaluation strategy for code editing. In Appendix E, the authors provide a justification for using GPT-4 for evaluation by comparing with human evaluation. However, I do not find this convincing for a few reasons. 1) The human evaluation is done by authors of this paper and not by external evaluators, and no information is given about annotator agreement, 2) The human evaluation entailed three classes (correct, partial, wrong), while GPT-4 evaluates based on 2 classes ("Yes" or "No"). For comparison, they group "partial" with "correct" while it should actually be considered wrong. 3) Finally, the consistency ratio is 68.4%, which is lower than what I would expect for strong evaluation.

**Reproducibility:**

4: Could mostly reproduce the results, but there may be some variation because of sample variance or minor variations in their interpretation of the protocol or method.

**Reviewer Confidence:**

5: Positive that my evaluation is correct. I read the paper very carefully and I am very familiar with related work.

**Typos Grammar Style And Presentation Improvements:**

- As mentioned in the Missing References, there is actually extensive work in code editing (e.g., CodeItT5, CodeReviewer) that could have been included as baselines in Table 2.

---

> ### Author Rebuttal · Authors · 2023-08-29
>
> Thank you for reviewing our paper and for the insightful comments. We hope our answers address your concerns about the paper.
>
> > Please address my first point above about using GitHub data directly or the CodeReviewer dataset.
>
>
> Thanks for the advice, we present additional experiments on training on Github data directly. Concretely, we employed the same automated procedure we utilized to collect or process the GitHub commits for our seed tasks. Through this, we collected 114K Github Python commits and their associated code snippets, which is the same size as CodeInstruct. These were derived from an expansive pool of nearly 3.5M commits.
>
> Given the limitations of time and resources, we performed our experiments with LLAMA-13B, and we report results below using varying fractions of the datasets respectively.
>
>
> | Data                   | GPT4_EVAL |
> |------------------------|-----------|
> | Github -50%            | 55.2      |
> | Github - 100%          | 62.7      |
> | CodeInstruct - 10%     | 67.7      |
> | CodeInstruct - 50%     | 73.6      |
> | CodeInstruct - 100%    | 75.9      |
>
> Upon evaluation, we found that the model, when fine-tuned with the CodeInstruct dataset, outperformed the equivalent model fine-tuned with GitHub data fetched directly. This superior performance was also observed when the model was fine-tuned with just 10% (roughly 11K data) of CodeInstruct, as it showed better results compared to using as much as 10 times of GitHub data.
>
> This evidence not only shows the superiority of the CodeInstruct dataset, but also shows the substantial effectiveness of prompt-based automatic data generation techniques described in the paper. These results validate our claim regarding the improved performance and reduced noise in our proposed dataset compared to direct GitHub examples.
>
> ---
>
> > The human evaluation is done by authors of this paper and not by external evaluators, and no information is given about annotator agreement
>
> We apologize for missing this important detail! The agreement rate among our annotators is 90.6%.
>
> > Finally, the consistency ratio is 68.4%, which is lower than what I would expect for strong evaluation.
>
> We adopted your suggestion that the “partial” class should be regarded as incorrect. As a result, the overall consistency is 0.8302, which is considered reasonably high. The evaluation of the finetuned LLAMA-33B model demonstrated the highest consistency with a score of 0.873.
>
>
> ---
>
> > One difference between code generation and code editing is that in editing, large portions of the input often get preserved. Have you explored output examples in some "diff" format which only requires generating the lines that actually changed?
>
> Your suggestion of utilizing a 'diff' format is indeed noteworthy as it could potentially reduce the number of generated tokens. However, the diff format is different from normal code, and the LLaMA model we extensively used throughout this paper was not pretrained on massive code diff data. [1] We made a conscious decision not to incorporate this approach due to its comparative complexity when contrasted with the direct generation of the complete edited code. Furthermore, its use introduces challenges in terms of diagnosis and debugging.
>
> [1] Touvron, Hugo, et al. "Llama: Open and efficient foundation language models." arXiv preprint arXiv:2302.13971 (2023).
>
> ---
>
> > For the seed data that was collected from GitHub, did you ensure that you used only license-permissive repositories? If not, using this dataset for anything beyond research purposes is likely not allowed, and this must be made clear. Please clarify this point.
>
> Yes, we ensured that all Github data we collected are from repositories with permissive licenses: MIT, Apache-2.0, GPL-3.0, GPL-2.0, BSD-2.0, BSD-3.0, LGPL-2.1, LGPL-3.0, AGPL-3.0.

---

### Official Review · Reviewer_H5fp · 2023-07-31

**Typos Grammar Style And Presentation Improvements:** line 177
**Soundness:** 3

**Excitement:**

4: Strong: This paper deepens the understanding of some phenomenon or lowers the barriers to an existing research direction.

**Missing References:**

None.

**Paper Topic And Main Contributions:**

This paper proposed a new benchmark (namely CodeInstruct) that evaluates various code edit tasks including comment insertion, code optimization, and code refactoring.
The dataset is augmented by utilizing GitHub corpus from BigQuery [1] as seed data, using ChatGPT [2].
Following [3], the main evaluation metric is done by GPT-4 [4], supported with human scoring.
The experimental results show that training LLMs with CodeInstruct is effective to improve their performance in the test set, e.g., the trained LLaMA-33B shows a comparable socre with ChatGPT (zero-shot).


[1] https://cloud.google.com/bigquery

[2] OpenAI. 2022. Introducing ChatGPT. https://openai.com/blog/chatgpt.

[3] Terry Yue Zhuo. 2023. Large language models are state-of-the-art evaluators of code generation. arXiv preprint arXiv:2304.14317.

[4] OpenAI. 2023. Gpt-4 technical report. https://arxiv.org/pdf/2303.08774.

**Questions For The Authors:**

* Can you report Cohen's kappa between GPT-4 and human decisions, and analyze the (dis)agreed categories?
* Can you report the category-wise performance, then analyze why LLMs good/bad at certain categories, and which categories that CodeInstruct is particularly effective along with conjectures?
* Can you build a hard test subset and report the GPT-4 and human metric scores for each model?

**Reasons To Accept:**

* Introduces a new dataset, following plausible steps to construct.
* Empirically shows the effectiveness of the benchmark by consistently improving the test accuracies (boh by GPT-4 and human) throughout LLMs.


**Reasons To Reject:**

* To justify using the automatic evaluation by GPT-4 as a main metric, authors need more supported analyses. I suggest Cohen's kappa between GPT-4 and human decisions, and analyze the (dis)agreed categories.
* (minor) This benchmark seems a bit easy-- ChatGPT Zero-shot: 90.5% by GPT-4 and 79.3% for correct scores by human (though it cannot be an apples-to-apples comparison, competition-level code generation like CodeContests by LLMs shows below 10% of Pass@1 (= the accuracy of generating a single code for each problem)). Can you analyze (and report) the category-wise performance, then build a hard subset for both 1) giving directions that what LLMs still bad at and 2) room for improvements that future work may acheive?

**Reproducibility:**

5: Could easily reproduce the results.

**Reviewer Confidence:**

4: Quite sure. I tried to check the important points carefully. It's unlikely, though conceivable, that I missed something that should affect my ratings.

---

> ### Author Rebuttal · Authors · 2023-08-29
>
> Thank you for the thoughtful review.
>
> > Can you report Cohen's kappa between GPT-4 and human decisions, and analyze the (dis)agreed categories?
>
> We classified “partially correct” from human evaluations as incorrect to maintain a binary categorization as in GPT-4 to compute the Cohen's kappa and the consistency (agreement) rate between GPT-4’s and humans’ evaluation. The overall Cohen's Kappa value is 0.665, with certain categories such as “Refactoring Code” and “Removing Dead Code” recording kappa values as 0.828 and 0.754 respectively, which suggests strong agreement. The overall consistency rate is 83%. Categories such as “Updating Library Dependencies” and “Refactoring Code” demonstrate high levels of agreement with consistency rate above 90% (93.8%/91.7%). However, the category of “Create Unit Tests” showed a lower consistency rate at 58.3% and kappa at 0.25. In summary, we observed that tasks characterized as “open-ended” tend to yield a low degree of agreement.
>
>
>
>
>
> > Can you report the category-wise performance, then analyze why LLMs are good/bad at certain categories, and which categories that CodeInstruct is particularly effective along with conjectures?
>
>
> We evaluated 2000 validation samples with GPT4 and presented the Top-10 Categories from Figure 2 in the following table.
>
>
> | Edit Intent categories          | 7B  | 13B  | 33B  |
> |----------------------------|------------------|-------------------|-------------------|
> | Add new features           | 0.749      | 0.812       | 0.892       |
> | Fix bugs                   | 0.634      | 0.683       | 0.79        |
> | Improve code readability   | 0.662      | 0.727       | 0.851       |
> | Error handling             | 0.89       | 0.939       | 0.933       |
> | General                    | 0.632      | 0.73        | 0.716       |
> | Code refactoring           | 0.66       | 0.738       | 0.786       |
> | Add logging/monitoring     | 0.811       | 0.911        | 0.94        |
> | Implement cache mechanism  | 0.779       | 0.792        | 0.896        |
> | Add comments/docstring     | 0.824       | 0.824        | 0.878        |
> | Create unit tests          | 0.79        | 0.887        | 0.887        |
>
>
> The categories “Add new features”, “Error handling”, and “Add logging/monitoring” demonstrate high performances in larger models (>0.9 in 33B). The LLMs perform well in these categories possibly due to the clarity of the rules of these tasks as they generally explicitly involve clear language instructions and semantics that are well processed by the model's language understanding abilities. In contrast, the “Fix bugs”, “General”, “Code Refactoring” categories had the lowest scores among the presented categories. “General” serves as a catch-all long-tail category which includes a wide array of potentially disparate and complex task instructions. These tasks are typically more complex as they often require the identification of subtlety in logic or implementation. Thus it requires a higher degree of reasoning abilities, which is challenging for current LLMs. We will add this analysis to the next revision of the paper.
>
>
> ---
>
> > Can you build a hard test subset and report the GPT-4 and human metric scores for each model?
>
> We managed to build a hard test subset which is constructed from test samples with an 'edit ratio' higher than 0.5, as described in Section 4.4. This specific threshold indicates that the respective model outputs needed considerable editing to conform to human-level reference standards.  This hard test subset has 67 examples in total  and the average edit ratio is 0.714. We report the results below.
>
> | Model | GPT4_EVAL | HUMAN_EVAL |
> | --- | --- | --- |
> | ChatGPT | 0.881 | 0.791 |
> | Ours (33B) | 0.866 | 0.731 |
> | Ous (13B) | 0.701 | 0.672 |
> | Ours (7B) | 0.597 | 0.507 |

---

### Official Review · Reviewer_XsAu · 2023-08-09

**Soundness:** 3
**Typos Grammar Style And Presentation Improvements:** 1. It would be better to provide exam…

**Excitement:**

4: Strong: This paper deepens the understanding of some phenomenon or lowers the barriers to an existing research direction.

**Missing References:**

1. Code edits dataset
  - Rosalia Tufano, Luca Pascarella, Michele Tufanoy, Denys Poshyvanykz, and Gabriele Bavota. 2021. Towards Automating Code Review Activities
 #### There are many bug fix dataset in software engineering area, e.g.
  - Yangruibo Ding, Baishakhi Ray, Premkumar Devanbu, and Vincent J Hellendoorn. 2020. Patching as Translation: the Data and the Metaphor.
   - https://github.com/rjust/defects4j
   - Michele Tufano, Jevgenija Pantiuchina, Cody Watson, Gabriele Bavota, and Denys Poshyvanyk. 2019. On learning meaningful code changes via neural machine translation

2. Code edit models
  - Jiyang Zhang, Sheena Panthaplackel, Pengyu Nie, Junyi Jessy Li, and Milos Gligoric. 2022. CoditT5: Pretraining for Source Code and Natural Language Editing.
  - CodeEditor: Learning to Edit Source Code with Pre-trained Models Jia Li, Ge Li, Zhuo Li, Zhi Jin, Xing Hu, Kechi Zhang, Zhiyi Fu
  - Yangruibo Ding, Baishakhi Ray, Premkumar Devanbu, and Vincent J Hellendoorn. 2020. Patching as Translation: the Data and the Metaphor.
  - Daniel Tarlow, Subhodeep Moitra, Andrew Rice, Zimin Chen, Pierre-Antoine Manzagol, Charles Sutton, and Edward Aftandilian. 2020. Learning to fix build errors with graph2diff neural networks
  - Changshu Liu, Pelin Cetin, Yogesh Patodia, Saikat Chakraborty, Yangruibo Ding, Baishakhi Ray, Automated Code Editing with Search-Generate-Modify
 - Jiayi Wei, Greg Durrett, Isil Dillig. 2023 Coeditor: Leveraging Contextual Changes for Multi-round Code Auto-editing

**Paper Topic And Main Contributions:**

## Summary
This paper introduced CodeInstruct, the first dataset designed to adapt LLMs for general-purpose code editing tasks. It consists of over 114K instruction-input-output triplets. The authors mined seed tasks from github commits and used them to inspire ChatGPT for generated instructions. They included various senarios together with instructions within prompt for ChatGPT to synthesize diverse input-output code snippets. The Authors showed that LLMs trained on CodeInstruct have comparable performance to ChatGPT.

## Contributions
1. First large-scale, diverse and general-purpose code-editing dataset
2. Novel approach to synthesize data from seed tasks on Github with the help of ChatGPT.
3. Their analysis revealed the high quality of the dataset.
4. The created dataset is effective in training a code editing LLM.

**Questions For The Authors:**

A. How do you filter the commit messages and finally classifiy them into 768 seed tasks? How do you manually create new seed tasks? What is the rule used here?
B. Have you thought about approaches to decrease the errors or noise generated by ChatGPT?
C. Did you manually clean each example in the test set to ensure they are reasonable and correct?
D. Why do you choose to only generate Python edits other than other programming languages?
E. Do you plan to run the fine-tuned models on other code-editing test data, automatic code review seems a reasonable task.
F. Do you think ChatGPT is able to generate test cases for edited code for evaluation?



**Reasons To Accept:**

1. I appreciate that the authors created the dataset used for fine-tuning LLM for code editing tasks. I believe this dataset will draw more attention and inspire further research in this area.
2. The dataset is built on the task seeds mined from github which give a more realistic starting point compared to other related dataset.
3. The analysis and plots show that the dataset covers diverse topics and intents. I like the analysis done by the authors.
4. The LLAMA-33B fine-tuned on this CodeInstruct dataset has compareble performance to ChatGPT.

**Reasons To Reject:**

1. The way to collect and filter the task seeds from git commits is not clearly described. I think this process is the most challenging part in collecting code-related data from Github.
   - Authors mentioned that "We used Codex to clarify the changes made between versions and improve the commit messages, resulting in more precise and informative instructions", but they did not described how they filter the commit messages and they did not provide any examples in this process.
   - I hope authors can provide details on how they classifiy the commit messages into 768 seed tasks. What is the criteria here. Is it simply by manually check?

2. My concern in evaluation is that there is only 134 data for evaluation which might be relatively small. Second, I would suggest evaluating on other out-of-box code editing dataset to gain more confidence about the quality of the dataset. (see missing related work for other datasets)
   - I am wondering if authors manually clean each example in the test set to ensure they are reasonable and correct. I think it is important for evaluate the models.
   - When comparing with Model fine-tuned on CodeAlpaca dataset, I am wondering if authors will also report the performance on test set of CodeAlpaca. Maybe the difference in performance is because of the data distribution shift.

3. The dataset only contains Python edits. It seems not hard to include other programming languages

**Reproducibility:**

4: Could mostly reproduce the results, but there may be some variation because of sample variance or minor variations in their interpretation of the protocol or method.

**Reviewer Confidence:**

4: Quite sure. I tried to check the important points carefully. It's unlikely, though conceivable, that I missed something that should affect my ratings.

---

> ### Author Rebuttal · Authors · 2023-08-29
>
> We thank reviewer XsAu for the careful review of our paper.
>
> > The way to collect and filter the task seeds from git commits is not clearly described.  How do you filter the commit messages and finally classify them into 768 seed tasks? How do you manually create new seed tasks? What is the rule used here?
>
> The filtering and collecting process is carefully designed with two main stages: automatic selection and manual curation. The automated steps effectively filter out the majority of unsuitable instructions. For the purpose of creating a dataset that better focuses on editing logic, we introduced the manual curation steps to obtain commits with clear and easy-to-identify edit intent.  We detail these two stages below.
>
>
> **Automatic**:
> * We selected github repos with over 100 stars to ensure the overall quality.
> * Repos with permissive license.(MIT, Apache-2.0, GPL-3.0, GPL-2.0, BSD-2.0, BSD-3.0, LGPL-2.1, LGPL-3.0, AGPL-3.0).
> * We kept commits in which only one single .py file was changed.
> * Using git-diff, we identified and preserved commits where only one code block was changed.
> * We discarded commits with a single-word or empty commit message.
> * Removed commits with edited rows number exceed 100.
>
>
>
> **Manual**:
> * Discarded commits containing only personal, project-specific logical adjustments that are irrelevant to the editing intent.
> * Discarded commits where the change in code doesn't match the commit message.
> * Removed commits involving considerable pseudocode, comments, debugging, or temporary changes.
> *  Utilized Codex for ambiguous commit messages, enhancing clarity in identifying code changes and intended edits.
>
> Through the automatic process, we initially obtained 6024 target commits data. We did not intentionally classify the seed tasks. The manual process further refined this data, ultimately narrowing down to a set of 768 ideal seed tasks. Both stages meld into a well-structured process that ensures our seed data is of high quality and clearly demonstrates the editing intent.
>
>
>
> ---
>
> > Have you thought about approaches to decrease the errors or noise generated by ChatGPT?
>
> Yes, in terms of using ChatGPT for data generation, we did occasionally observe errors or noises. We have indeed undertaken steps to mitigate such errors, which we would like to outline below:
>
> * Detection and removal of failed generation attempts: We noticed instances where ChatGPT failed or refused to generate input-output code pairs. We employed a keyword-based filtering method in which we created a set of specific keyword-matching rules, based on terms such as “Sorry”, “I cannot” and similar expressions that are typically indicative of a failed generation.
> * Prevention of redundant data: We initially found a few instances where the input already fulfilled the intent of the editing instruction. We removed instances that had the same input and output. We also iteratively refined our prompts for generating the data to reduce such cases, and finally consolidated to the prompt disclosed.
>
>
> ---
>
> > Did you manually clean each example in the test set to ensure they are reasonable and correct?
>
> Yes,  we conducted a careful manual review and cleanup to ensure all instances were rational and accurate. This effort led to the retention of 134 examples in our final, quality-assured test set.
>
> ---
>
> > Why do you choose to only generate Python edits other than other programming languages?
>
> We want to note that our data generation pipeline is intrinsically language-agnostic, and is capable of focusing on specific topics, which holds potential benefits for downstream tasks. The decision to use Python as the dedicated language for code edits was driven by multiple considerations:
>
> * Python has vast and professional repositories, which ensures the quality of our seed data.
>
> * Our language choice was influenced by the familiarity and expertise of our research team. This decision ensures high accuracy and integrity in our manually curated seed tasks.
>
>  We selected Python as an example language to validate our methodology's effectiveness, and we leave the extension of our method to other programming languages as future work.
>
>
> ---
>
> > Do you plan to run the fine-tuned models on other code-editing test data, automatic code review seems a reasonable task
>
> We agree that running fine-tuned models on other code-editing test data represents an important step in validating the versatility of our model. Automatic code review indeed presents an interesting avenue of application, but they do not perfectly align with our purpose of creating the dataset.
> Our CodeInstruct dataset, while unique in its structure and focus on instruction fine-tuning on a variety of code editing tasks, currently consists of only Python data. We acknowledge that this limitation affects the direct comparability of our work with other datasets like CodeReviewer[1], Defect4j[2], and those referred to in [3] and [4]. CodeAlpaca[5] does not have a test set. These datasets, while insightful, feature a variety of programming languages that our current fine-tuned model is not equipped to handle.
>
>
>
> [1] Li, Zhiyu, et al. "Automating code review activities by large-scale pre-training." Proceedings of the 30th ACM Joint European Software Engineering Conference and Symposium on the Foundations of Software Engineering. 2022.
>
> [2] Just, René, Darioush Jalali, and Michael D. Ernst. "Defects4J: A database of existing faults to enable controlled testing studies for Java programs." Proceedings of the 2014 international symposium on software testing and analysis. 2014.
>
> [3]Tufano, Michele, et al. "On learning meaningful code changes via neural machine translation." 2019 IEEE/ACM 41st International Conference on Software Engineering (ICSE). IEEE, 2019.
>
> [4] Ding, Yangruibo, et al. "Patching as translation: the data and the metaphor." Proceedings of the 35th IEEE/ACM International Conference on Automated Software Engineering. 2020.
>
> [5] https://github.com/sahil280114/codealpaca
>
> ---
>
> > Do you think ChatGPT is able to generate test cases for edited code for evaluation?
>
> We have considered using ChatGPT to generate test cases for edited code. However, in practice, we found that the feasibility to generate executable test cases are somewhat task-dependent.
>
> For example, while ChatGPT is capable of generating unit test cases for straightforward tasks like “Error Handling” or “Update Library Version”, it falls short of generating an executable test case for tasks like, “Generate unit-test”, “Code Refactoring”, etc. Such tasks often demand specific, task-dependent engineering for executing and verifying the generated test case.
>
> ---
>
> >1.  It would be better to provide examples of different edit intents in the appendix.
> >2.    The demo figure in Supplementary Materials should be added to paper
>
>
> Thank you for your advice. We will provide examples from the dataset with different edit intents, and add a clear demo figure to the paper.

---

### Meta-Review · Area_Chair_U7Bt · 2023-09-01

**Recommendation:** Reject
**Confidence:** 5

**Metareview:**

The authors create a very interesting instruction dataset combining real GitHub data with ChatGPT expanded samples. While the discussion and motivation of the dataset creation method is very interesting and well-describe, the only evaluations they offer to demonstrate the value of this finetuning dataset are automatic GPT-4 based evaluations. The authors justify this criterion by citing a paper claiming that LLMs are effective at evaluating the quality of generated code, but this paper only compares the LLM generated evaluations to sequence similarity based metrics which are extremely problematic especially for code generation. See for example arxiv:2201.12901 figure 7 and arXiv:2108.07732 figure 10. At a minimum, showing a correlation between the automatic LLM evaluation and a code execution-based evaluation like HumanEval, MBPP, DataScienceProblems would be necessary to make a claim that LLM-based evaluation is effective. Even so, such evidence is only indirect and is not sufficient to make empirical claims.

---

### Decision · Program_Chairs · 2023-10-07

**Decision:**

Reject

**Comment:**

The authors create a very interesting instruction dataset combining real GitHub data with ChatGPT expanded samples. While the discussion and motivation of the dataset creation method is very interesting and well-describe, the only evaluations they offer to demonstrate the value of this finetuning dataset are automatic GPT-4 based evaluations. The authors justify this criterion by citing a paper claiming that LLMs are effective at evaluating the quality of generated code, but this paper only compares the LLM generated evaluations to sequence similarity based metrics which are extremely problematic especially for code generation. See for example arxiv:2201.12901 figure 7 and arXiv:2108.07732 figure 10. At a minimum, showing a correlation between the automatic LLM evaluation and a code execution-based evaluation like HumanEval, MBPP, DataScienceProblems would be necessary to make a claim that LLM-based evaluation is effective. Even so, such evidence is only indirect and is not sufficient to make empirical claims.